# Age-dependent neuroprotective effect of an SK3 channel agonist on excitotoxity to dopaminergic neurons in organotypic culture

Oscar Maldonado[1☯], Alexandra Jenkins[2☯], Helen M. Belalcazar[3], Helena Hernandez-Cuervo[2], Katelynn M. Hyman[2], Giannina Ladaga📷[1], Lucia Padilla[1], Gabriel A. de Erausquin📷[1]*

1 Laboratory of Brain Development, and Repair, Biggs Institute for Alzheimer and Neurodegenerative Diseases, University of Texas Health San Antonio, San Antonio, Texas, United States of America,
2 Department of Molecular Medicine, University of South Florida, Tampa, Florida, United States of America,
3 Department of Genetics, Albert Einstein College of Medicine, New York City, New York, United States of America

☯ These authors contributed equally to this work.
* deerausquing@uthscsa.edu

**Data Availability Statement:** All relevant data are included within the manuscript.

## Abstract

### Background

Small conductance, calcium-activated (SK3) potassium channels control the intrinsic excitability of dopaminergic neurons (DN) in the midbrain and modulate their susceptibility to toxic insults during development.

### Methods

We evaluated the age-dependency of the neuroprotective effect of an SK3 agonist, 1-Ethyl-1,3-dihydro-2H-benzimidazol-2-one (1-EBIO), on Amino-3-hydroxy-5-methylisoxazole-4-propionic acid (AMPA) excitotoxicity to DN in ventral mesencephalon (VM) organotypic cultures.

### Results

Most tyrosine hydroxylase (TH)+ neurons were also SK3+; SK3+/TH- cells (DN+) were common at each developmental stage but more prominently at day in vitro (DIV) 8. Young DN+ neurons were small bipolar and fusiform, whereas mature ones were large and multipolar. Exposure of organotypic cultures to AMPA (100 μm, 16 h) had no effect on the survival of DN+ at DIV 8, but caused significant toxicity at DIV 15 (n = 15, p = 0.005) and DIV 22 (n = 15, p<0.001). These results indicate that susceptibility of DN to AMPA excitotoxicity is developmental stage-dependent in embryonic VM organotypic cultures. Immature DN+ (small, bipolar) were increased after AMPA (100 μm, 16 h) at DIV 8, at the expense of the number of differentiated (large, multipolar) DN+ (p = 0.039). This effect was larger at DIV 15 (p<<0.0001) and at DIV 22 (p<<0.0001). At DIV 8, 30 μM 1-EBIO resulted in a large increase in DN+. At DIV 15, AMPA toxicity was prevented by exposure to 30 μM, but not

**Funding:** This work was supported in part by the Zachry Foundation Chair to Gde, and the Center for Brain Health grant from the Valley Baptist Legacy Foundation to GdE. In addition, the authors received funding from Roskamp Chair at USF Morsani College of Medicine and Fundacion FULTRA. The funders had no role in study design, data collection and analysis, decision to publish, or preparation of the manuscript.

**Competing interests:** The authors have declared that no competing interests exist.

**Abbreviations:** VM, ventral mesencephalon; 1-EBIO, 1-Ethyl-1,3-dihydro-2H-benzimidazol-2-one; 1-Ethylbenzimidazolinone; CyPPA, Cyclohexyl-[2-(3,5-dimethyl-pyrazol-1-yl)-6-methyl-pyrimidin-4-yl]-amine; PBS, phosphate buffered saline; AMPA, Amino-3-hydroxy-5-methylisoxazole-4-propionic acid; NMDAR, N-methyl-D-Aspartate type glutamate receptor; TH, tyrosine hydroxylase; SK, small conductance calcium-activated potassium channel; SK3, small conductance calcium-activated potassium channel 3 also known as $K_{Ca}2.3$; DIV, days in vitro; DN, dopaminergic neurons; KCNN3, potassium calcium-activated channel subfamily N member 3 (human) gene; DMSO, Dimethyl Sulfoxide; NGS, normal goat serum; DAB, 3,3′-Diaminobenzidine; HRP, horseradish peroxidase; DAPI, 4′,6-diamidino-2-phenylindole; ROI, region of interest.

100 μM 1-EBIO. At DIV 22, excitotoxicity was unaffected by 30 μM 1-EBIO, and partially reduced by 100 μM 1-EBIO.

## Conclusion

The effects of the SK3 channel agonist 1-EBIO on the survival of SK3-expressing dopaminergic neurons were concentration-dependent and influenced by neuronal developmental stage.

## Introduction

The intrinsic excitability and suceptiblity to toxic inuslts during development of dopaminergic neurons (DN) in the midbrain are regulated by small conductance, calcium-activated potassium (SK3) channels [1–3]. Abnormalities in SK3 channel function may contribute to the risk of schizophrenia, as mutations in the *KCNN3* gene are associated with dopamine neuron dysfunction [4, 5] as well as with risk of schizophrenia in some patient samples [6–9]. Developmental processes in dopaminergic neurons (DN) have been extensively studied and characterized using primary cultures of embryonic mesencephalon [10, 11]. Indeed, the signaling pathways that define neuronal identity, maturation and differentiation of DN were described in primary cultures [12]. Furthermore, DN in cultures show mature functional phenotypes [13–17]. Our previous work in primary cultures firmly established that protracted stimulation of glutamate receptors activated by α-amino-3-hydroxy-5-methyl-4-isoxazolepropionic acid (AMPA) results in phenotype-specific toxicity to a subpopulation of DN [18–21]. By contrast, we found that direct application of the glutamate receptor agonist N-Methyl-d-aspartate (NMDA) is not toxic to cultured DN [20], and fails to induce significant changes in intracellular free calcium concentrations or increase phosphorylation of c-AMP responsive element binding protein (CREB) in the same system [18]. Thus, rather than an accurate model of excitotoxicity to DN in neurodegenerative diseases such as Parkinson's disease, AMPA-induced death of DN appears to mimic natural cell death of DN occurring during neurodevelopment [22] and the mechanism by which susceptible populations are affected in psychiatric neurodevelopmental disorders such as schizophrenia [23]. Indeed, primary cultures are plated during a critical period of the ontogeny of excitatory glutamatergic circuitry between the subthalamic nucleus and the substantia nigra pars compacta, in which AMPA receptors reach the peak of expression at the midbrain [24, 25].

In this model, commitment to die requires previous suppression of SK3 channel activity, and pharmacological enhancement of SK3 conductivity results in neuroprotection of DN [3]. We also found that a similar cell death mechanism can be triggered in developing midbrain in utero by maternal infection with neurotropic influenza virus, [18] and proposed that the same molecular pathways may act as a putative mechanism for the loss of neurons in the mesocortical dopaminergic projection in abnormal neurodevelopment leading to negative symptoms in schizophrenia [23]. However, primary cultures are limited in their viability in vitro and only allow observation of short-term effects of toxicity [26]. Organotypic cultures retain some of the 3D shape and some modicum of connections with normal anatomical targets, and most importantly, can be maintained in culture for much longer periods of time [26]. This report describes age-dependent susceptibility to excitotoxity of dopaminergic neurons in organotypic culture, as well as the impact of neuroprotection by an SK3 agonist, 1-EBIO, in organotypic cultures of different ages.

## Materials and methods

### Animals and dissections

Organotypic cultures were established from the ventral mesencephalon (VM) of rat embryos. Timed pregnant female Sprague-Dawley rats (n = 30; Charles River Laboratories, MA, USA) were euthanized by exposure to $CO_2$ on day 14 of gestation. Following laparotomy, embryos were quickly removed and placed in ice-cold Hanks Balanced Salt Solution (Sigma-Aldrich, MO, USA). Brains were dissected, VMs isolated, and meninges carefully removed [27] using a stereoscope (Zeiss, Oberkochen, Germany). VMs used in toxicological stereology experiments were carefully sliced at the midline for treatment and control sections. Animals were treated in accordance with the National Institutes of Health (NIH) Guide for the Care and Use of Laboratory Animals. The University of South Florida Institutional Animal Care and Use Committee approved all procedures (protocol IS00000427).

### Organotypic cultures

Dissected VMs were cultured following established procedures [28]. Briefly, VMs were transferred to Millicell-CM inserts (EMD Millipore, MA, USA) in six-well plates with 2 VM hemisections on each insert. Each well received 1.2 ml plating medium containing Neurobasal medium (Invitrogen, MA, USA) supplemented with 20% horse serum (Sigma-Aldrich, MO, USA), B-27® (50X, Invitrogen, MA, USA), Glutamax™ (100X, Invitrogen, MA, USA), D-(+)-glucose (1M, Sigma-Aldrich, MO, USA), penicillin-streptomycin (100X, Sigma-Aldrich, MO, USA), and basic fibroblast growth factor (25 µg/ml, Sigma-Aldrich, MO, USA). Differentiation was induced after 5 days *in vitro* (DIV) by introducing culture medium containing Neurobasal medium (Invitrogen, MA, USA) supplemented with 1% horse serum (Sigma-Aldrich, MO, USA), B-27® (50X, Invitrogen, MA, USA), Glutamax™ (100X, Invitrogen, MA, USA), and penicillin-streptomycin (100X, Sigma-Aldrich, MO, USA). Culture medium was renewed every 48 hours until toxicity experiments were performed on DIV 7, 14 and 21. Cultures were maintained at 37˚C in an atmosphere of 5% $CO_2$ and 100% relative humidity.

### Pharmacological treatments

All drug concentrations were chosen based on recommendations from manufacturers or previously published data by our own and other relevant groups [3, 18]. For DN survival experiments cultures were exposed to alpha-amino-3-hydroxy-5-methyl-4-isoxazolepropionic-acid (AMPA) (100 µM) for 24 hours (DIV 7, 14, 21). Treatment was stopped by briefly rinsing with phosphate buffered saline (PBS) (Sigma-Aldrich, MO, USA), followed by fixation (DIV8, DIV15, DIV22). Cultures used for SK3 channel activation experiments were exposed to 1-ethyl-2-benzimidazolinone (1-EBIO) (30, 100 µM), and (AMPA) (100 µM) for 16 hours (DIV 7, 14, 21) in the following treatment groups: 1-EBIO 30 µM, 1-EBIO 100 µM, 1-EBIO 30 µM plus AMPA 100 µM, 1-EBIO 100 µM plus AMPA 100 µM, and AMPA 100 µM. Treatments were stopped by rinsing briefly with PBS, followed by fixation (DIV 8, DIV 15, DIV 22) for TH and SK3 immunofluorescence. 1-EBIO was dissolved in DMSO (Sigma-Aldrich, MO, USA) and AMPA in double distilled, deionized water as indicated in the corresponding product sheets. Both pharmacological agents were obtained from Tocris Bioscience, MO, USA.

### Immunohistochemistry

Cultures were fixed in 4% formaldehyde (Fisher Scientific, MA, USA)– 4% sucrose (Sigma-Aldrich, MO, USA) in PBS at 4˚C for 24 hours. Cultures were then sliced at a thickness of 50 µm using a Vibratome V1200 (Leica Biosystems, Wetzlar, Germany) and mounted on

Fisherbrand™ Superfrost™ microscope slides. Mounted sections were permeabilized with 0.2% Triton (Sigma-Aldrich, MO, USA) for 15 minutes and blocked for endogenous peroxidase with 3% $H_2O_2$ (Sigma-Aldrich, MO, USA) for 10 minutes, followed by blocking with 10% normal goat serum (NGS) (Sigma-Aldrich, MO, USA) for 30 minutes. Sections were then incubated with the primary antibody mouse anti-TH (1:1000 EMD Millipore, MA, USA) for one hour at room temperature, followed by incubation with a goat anti-mouse IgG (H+L) poly-HRP secondary antibody (1:250 Invitrogen, MA, USA) for one hour at room temperature. For each antibody, tissue sections were incubated with primary antibody alone to test for auto-fluorescence, and with the secondary antibody alone to test for non-specific binding, at the same concentrations and in the same conditions used for the results; non-specific staining and autofluorescence were negligible (data not shown), For visualization, sections were then incubated with SIGMA FAST DAB (3,3'-Diaminobenzidine) per manufacturer's instructions (Sigma-Aldrich, MO, USA), followed by a brief washing with PBS to stop the reaction. Tissue sections were then dehydrated and cleared using increasing concentrations of ethanol and xylene, followed by coverslip application using Crystal mount ™ aqueous mounting medium (Sigma-Aldrich, MO, USA). All antibody solutions were made using 1% NGS in PBS.

## Immunofluorescence

Cultures were fixed in 4% formaldehyde—4% sucrose in PBS at 4˚C for 24 hours. Cultures were then sliced at a thickness of 30 μm using a Vibratome V1200 (Leica Biosystems, Wetzlar, Germany) and mounted on Fisherbrand™ Superfrost™ microscope slides. Mounted sections were permeabilized in 0.15% Tween-20/0.05% Triton (Sigma-Aldrich, MO, USA) for 15 minutes and blocked in 10% normal goat serum (NGS) (Sigma-Aldrich, MO, USA) for 30 minutes, followed by blocking for endogenous avidin and biotin (avidin/biotin blocking kit, Invitrogen, MA, USA) per manufacturer's instructions. Sections were then incubated with the primary antibodies mouse anti-TH (1:1000 EMD Millipore, MA, USA) and rabbit anti-SK3 (1:1,500 Alomone labs, Jerusalem, Israel) for 20 hours at 4˚C followed by incubation with biotinylated goat anti-rabbit IgG (H+L) (Abcam, Cambridge, UK) for 15 minutes. Sections were incubated with the secondary antibodies goat anti-mouse Alexa Fluor® 568 (1:500) and Alexa Fluor® 488 conjugated to streptavidin (1:500) (Invitrogen, MA, USA) for 2 hours at room temperature. Sections were rinsed three times with PBS and cover-slipped using Prolong® Gold anti-fade reagent with DAPI (Invitrogen, MA, USA). All antibody solutions were made using 1% NGS/0.02% each Tween-20 and Triton in PBS.

## Stereology and assessment of DN survival

Every other section of VM was stained with TH for unbiased stereological cell counting using the optical fractionator method [29]. The total number of TH+ neurons were counted in VM's (DIV 8, DIV 15, DIV 22) (n = 14–15 toxicity experiments per DIV) by individuals blind to treatment conditions. Counts of TH+ neurons within a counting frame (50% screen height$^2$) were made at regular predetermined intervals (frame spacing = 150 μm) by means of a grid program (Stereologer 2000, Stereology Resource Center, Inc., FL, USA). Each VM was viewed at low magnification (Nikon Apo 4x dry objective) and the entire section was outlined as the region of interest (ROI). Then, at a systematically determined starting point, TH+ neurons were counted at high magnification (Nikon Apo 60x oil objective). Cells were counted as TH + if they exhibited immunoreactivity in the soma and the dendrites, and if the nucleus of the cell was within the focus between the top and bottom boundaries of the counting frame without touching the exclusion lines. Neurons were differentiated from non-neuronal cells based on size, shape and lack of TH staining. After counting the total number of TH+ neurons in

each section, the program generated an estimate of the total number of TH+ neurons using the sample sectioning fraction (SSF), the area sampling fraction (ASF), and the mean thickness.

## Effects of 1-EBIO on DN survival

One in three sections of control and 1-EBIO/AMPA treated VM were stained with TH and SK3, and were used for unbiased stereological cell counting (Stereologer, 2000, Stereology Resource Center, Inc., FL, USA). The total number of TH+/SK3+ co-stained neurons were counted in VM's (DIV 8, DIV 15, DIV 22) (n = 3 toxicity experiments per DIV) using the optical fractionator probe in a blind-treated manner. Counts of TH+/SK3+ neurons within a counting frame (50% screen height$^2$) were made at regular predetermined intervals (frame spacing = 100 μm) by means of a grid program (Stereologer 2000, Stereology Resource Center, Inc., FL, USA). Stereological estimates for total number of TH+ neurons were calculated as described above. Estimates of total number of SK3+ neurons were calculated manually using the fractionator equation as described by Peter Mouton [29]. All stereological counting was carried out using a Nikon Diaphot 300 inverted microscope (Nikon, Tokyo, Japan) equipped with a stereologer system (Stereologer 2000, Stereology Resource Center, Inc., FL, USA), 4x dry and 60x oil immersion objectives, 75 W xenon lamp, rhodamine and fluorescein filters, motorized XYZ stage (Stereology Resource Center, Inc., FL, USA), and Panasonic industrial CDD Gp-KR22 digital camera (Panasonic, Kadoma, Japan).

## Measurement of soma size

Digital images of microscope fields containing at least one DN were captured, and DN soma diameter was measured at the widest point using ImageJ (NIH, MD, USA). Soma diameter was measured in pixels and converted to μm using a known measurement to set the scale within ImageJ.

## Statistical analysis

Data were analyzed using SPSS (version 22 for Macintosh, IBM, USA). Results are reported as mean ± standard error. For DN survival at least 14 AMPA toxicity replicates were performed for each DIV. Statistical analysis was performed by two-way ANOVA with DIV and treatment as fixed factors, followed by protected Post-hoc Test for multiple comparisons. For 1-EBIO experiments at least 3 replicates were performed for each treatment group. Statistical analysis was performed by one-way ANOVA on each DIV, with treatment as the fixed factor, followed (in significant ANOVAS) by simple contrasts to compare each treatment group to the control. The alpha level for significance was set at $P \leq 0.05$. Prior to analysis all data was checked for normality using the Shapiro-Wilk normality test, and checked for outliers using the outlier-labeling rule [30].

## Results

### SK3 channels are expressed in DN organotypic cultures

Virtually all TH+ neurons observed were also SK3+. Only occasionally did we observe neurons that were TH+/SK3-, most frequently at DIV 8. SK3 staining was mainly localized to the soma and proximal dendrites, with staining being more intense closer to the cell body (Fig 1, panel A). SK3+/TH- cells were common at each developmental stage but more prominently at DIV8, which contrasts with our results using primary cultures. TH+/SK3+ neurons exhibited two distinctive morphologies: small to medium bipolar fusiform and large multipolar. Most

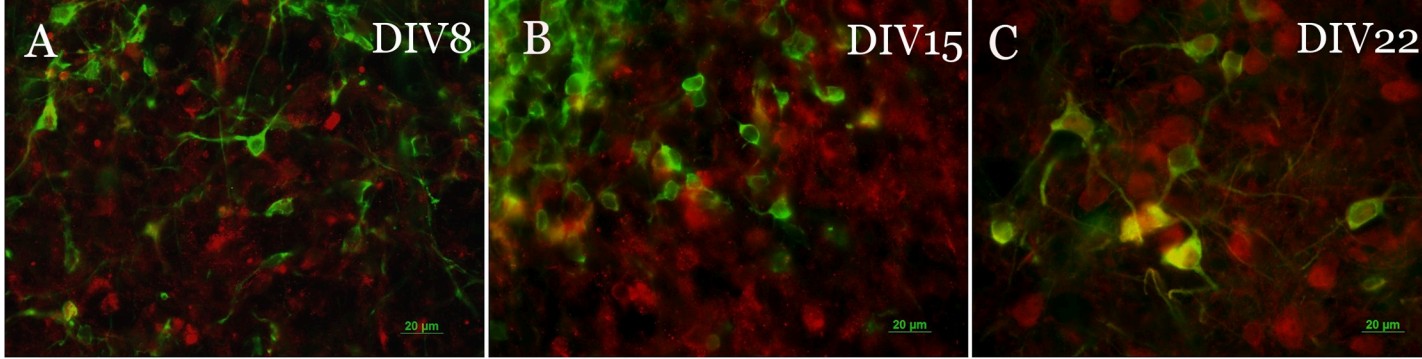

**Fig 1. Double immunofluorescence labeling of embryonic rat VM organotypic cultures with antibodies against TH and SK3.** Images show examples of stained neurons at DIV 8 (panel A), DIV 15 (panel B) and DIV 22 (panel C) merged color images show high co-localization for anti-TH (red) and anti-SK3 (green) antibodies. At each DIV there are also a number of cells that are positive for anti-SK3, but negative for anti-TH. At DIV 8 DNs were typically bipolar with small fusiform shaped somas and displayed little branching. While small bipolar neurons were still present at DIV 15, DN morphology began shifting towards multipolar cells with larger somas and increased branching. Multipolar cells with large somas was the predominant DN morphology at DIV 22, with small bipolar cells being rare. Scale bar = 20 μm.

TH+/SK3+ neurons at DIV 8 were small, spherical, and bipolar, whereas at DIV 22 TH+/SK3+ they were frequently large and multipolar (Fig 1, panels A, C).

## DNs in organotypic cultures are differentially susceptible to AMPA-receptor mediated excitotoxicity

AMPA-receptor mediated excitotoxicity was assessed in organotypic VM cultures at DIV 8, DIV 15, and DIV 22. After 16 hours of exposure to AMPA (100 μm), DIV 8 treated tissue did not exhibit a detectable loss of TH+ cells compared to control tissue (CT) (n = 14, F = 0.124; $p = 0.127$, Fig 2, panel A and top row of panel B). At DIV 15, however, AMPA treatment

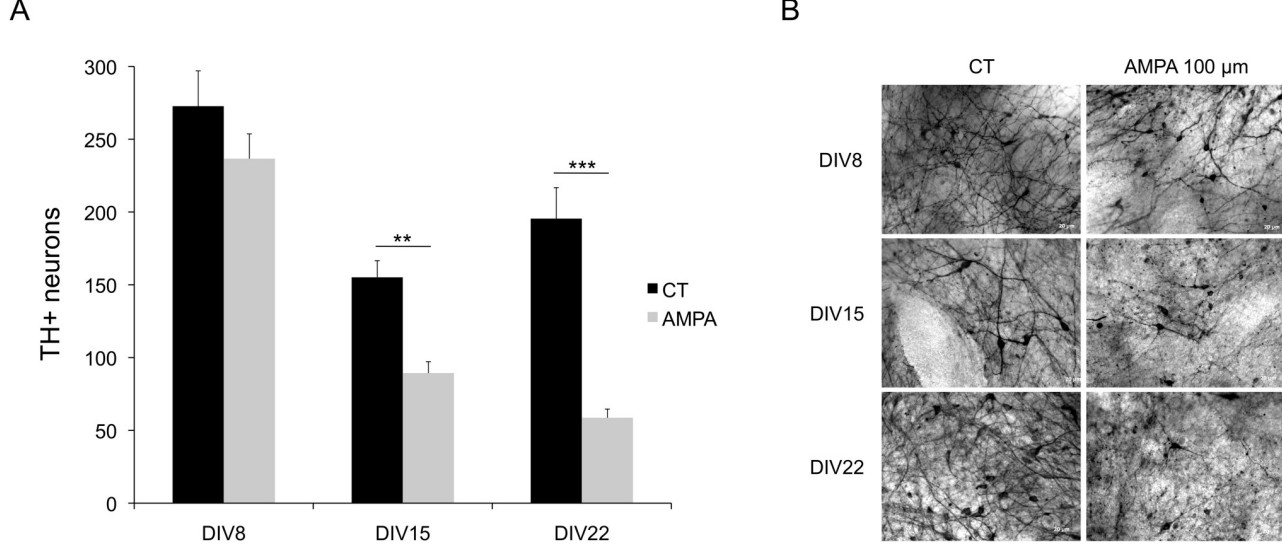

**Fig 2. Susceptibility to AMPA-induced excitotoxicity is developmental stage dependent in organotypic cultures of embryonic VM.** (A) VM were incubated with AMPA (100 μM) for 16 hr at DIV 8 (n = 14), DIV 15 (n = 15), and DIV 22 (n = 15). Survival is expressed as total number of TH + neurons in untreated VM (CT) compared to VM treated with AMPA (100 μM) for each DIV. Data are expressed as mean ± SEM. **$p < 0.01$, ***$p < 0.001$. (B) Representative photomicrographs of DN at DIV 8, DIV 15 and DIV 22 comparing untreated (CT) VM and VM treated with AMPA (100 μM) for 16 hr. Scale bar = 20 μm.

(100 μm, 16 hr) showed a significant decrease in the number of TH+ cells compared to CT tissue (n = 15, F = 10.826; $p<0.005$, Fig 2 panel A and middle row of panel B). The most dramatic effect of AMPA treatment (100 μm, 16 hr) on DN survival was observed at DIV 22, where the frequency of TH+ cells was diminished by more than half (n = 15, F = 21.651, $p<0.0001$, Fig 2, panel A and bottom row of panel B). These results indicate that susceptibility of DN to AMPA excitotoxicity is developmental stage-dependent in embryonic VM organotypic cultures.

Since the most notable change in the appearance of DN with time in vitro was their soma size, we also measured the soma diameter of individual DNs in AMPA-treated and CT tissue at each DIV to establish the impact of differentiation on susceptibility. Interestingly, although we did not detect an effect of AMPA treatment on overall DN survival at DIV 8, the number of large DN was significantly lower at DIV 8 compared to CT tissue ($n_{CT} = 163$, $n_{AMPA} = 126$, $p = 0.039$, Fig 3, panel B). This pattern was most dramatic at DIV 15 ($n_{CT} = 151$, $n_{AMPA} = 146$, $p = 2.0 \times 10^{-24}$, Fig 3, panel B) and continued to be significant at DIV 22 ($n_{CT} = 133$, $n_{AMPA} = 111$, $p = 1.7 \times 10^{-10}$, Fig 3, panel B). These results indicate that AMPA treatment has a differential effect on more differentiated DN in embryonic VM organotypic culture.

## Protection conferred by SK3 channel activation is developmental stage-dependent in organotypic cultures of embryonic VM

The effects of the SK3 channel agonist 1-EBIO on the survival of SK3-expressing dopaminergic neurons were concentration-dependent and influenced by developmental age (Fig 4, two-way ANOVA F = 36.242, p < 0.00001). At DIV 8 there was no significant toxicity of AMPA, but treatment with 30 μM resulted in a large increase in double stained neurons (Fig 4, panel A). At DIV 15, AMPA toxicity was prevented by exposure to 30 μM, but not 100 μM 1-EBIO (Fig 4, panel B). Lastly, at DIV 22, the profound excitotoxic effect of AMPA was unaffected by 30 μM 1-EBIO, and only partially reduced by exposure to 100 μM 1-EBIO (Fig 4, panel C).

## Discussion

Dopamine neuron susceptibility to excitotoxic injury is potentially relevant to a number of pathological conditions, and therefore so is the development of potential neuroprotective strategies, including the use of pharmacological modulators of SK3 channel function [31–34]. We previously showed that 1-EBIO effectively protects DN in primary cultures of ventral mesencephalon against AMPA-mediated toxicity by sustaining SK3 channel conductance [3]. Specifically, we found that the inactivation of SK3 channels by apamin or NS8593 promotes specific death of DN sharing a convergent pathway with AMPA-induced excitotoxicity [3]. By contrast, the SK3 channel agonists 1-EBIO and CyPPA increased SK3 currents and promoted both survival and differentiation of cultured DN, as well as increased neurogenesis [3, 35]. The overall expression level of SK3 channel increases with the synaptic maturation [36]. The trophic effects of SK3 agonists are likely explained by the rescue of DN that degenerate as a result of culture-related neuronal attrition [3]. However, primary cultures can be studied over a very brief period of time, and lack much of the natural geometry of the brain tissue including important interactions with glial cells and other neurons. Thus, we studied organotypic cultures of the ventral mesencephalon in the same species and were harvested at the same age, but were maintained in vitro long enough to allow evaluation of DN susceptibility at different developmental stages up to 22 days in vitro.

Perhaps not surprisingly, we found that in organotypic cultures, susceptibility of DN to AMPA-mediated excitotoxicity increases with age and differentiation status, such that the more complex, larger DN that become more common as the tissue matures, are much more susceptible (Fig 3, panel A) than the embryonic-appearing, small and round neurons more common earlier in development (Fig 3, panel B). This is consistent with observations where

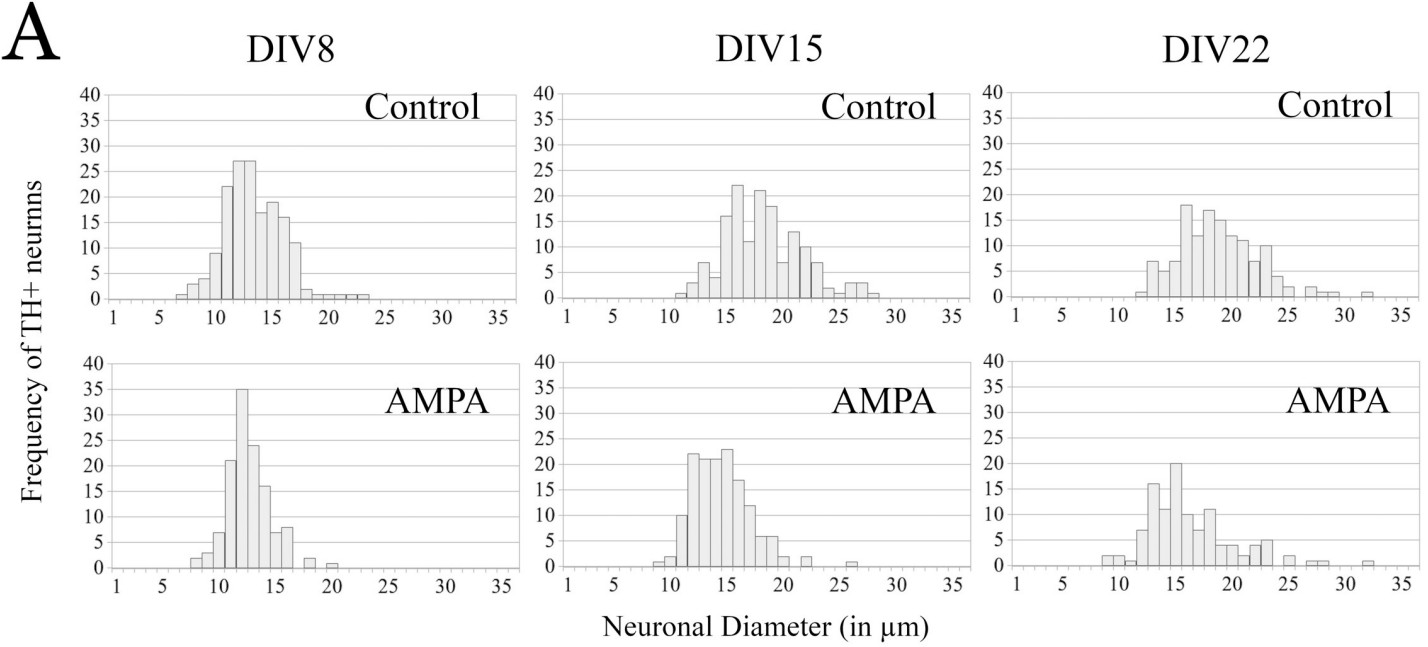

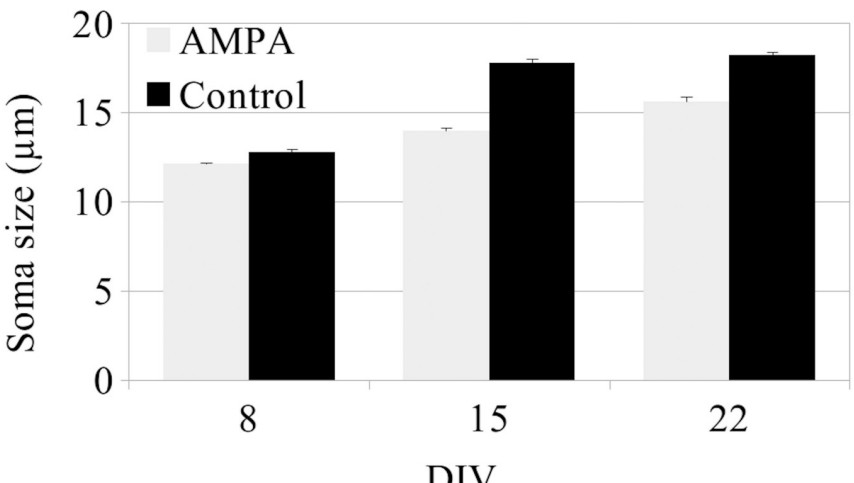

**Fig 3. AMPA treatment preferentially targets DN with larger soma size in embryonic VM organotypic cultures.** (A) Histogram representing the frequency of DN soma size in untreated (CT) VM and VM treated with AMPA (100 μM) for 16 hr at DIV 8 ($n_{CT}$ = 126, $n_{AMPA}$ = 163), DIV 15 ($n_{CT}$ = 151, $n_{AMPA}$ = 146) and DIV 22 ($n_{CT}$ = 133, $n_{AMPA}$ = 111). (B) Mean DN soma size following 16 hr AMPA treatment in DIV 8, DIV 15, and DIV 22. Soma size is expressed as mean DN diameter (μm) ± SEM for all DIV.

age plays a determining role in the increased susceptibility of DN to chronic rotenone exposure that is accompanied by severe locomotor deficits and decreased lifespan [37]. Overall, AMPA did not induce cell death in DN at 8 days in vitro, but toxicity became apparent at 15 DIV and most intense at 22 DIV. These findings are consistent with knowledge about the expression of AMPA receptors in DN in VM or rodents [3]. Activity-dependent changes in

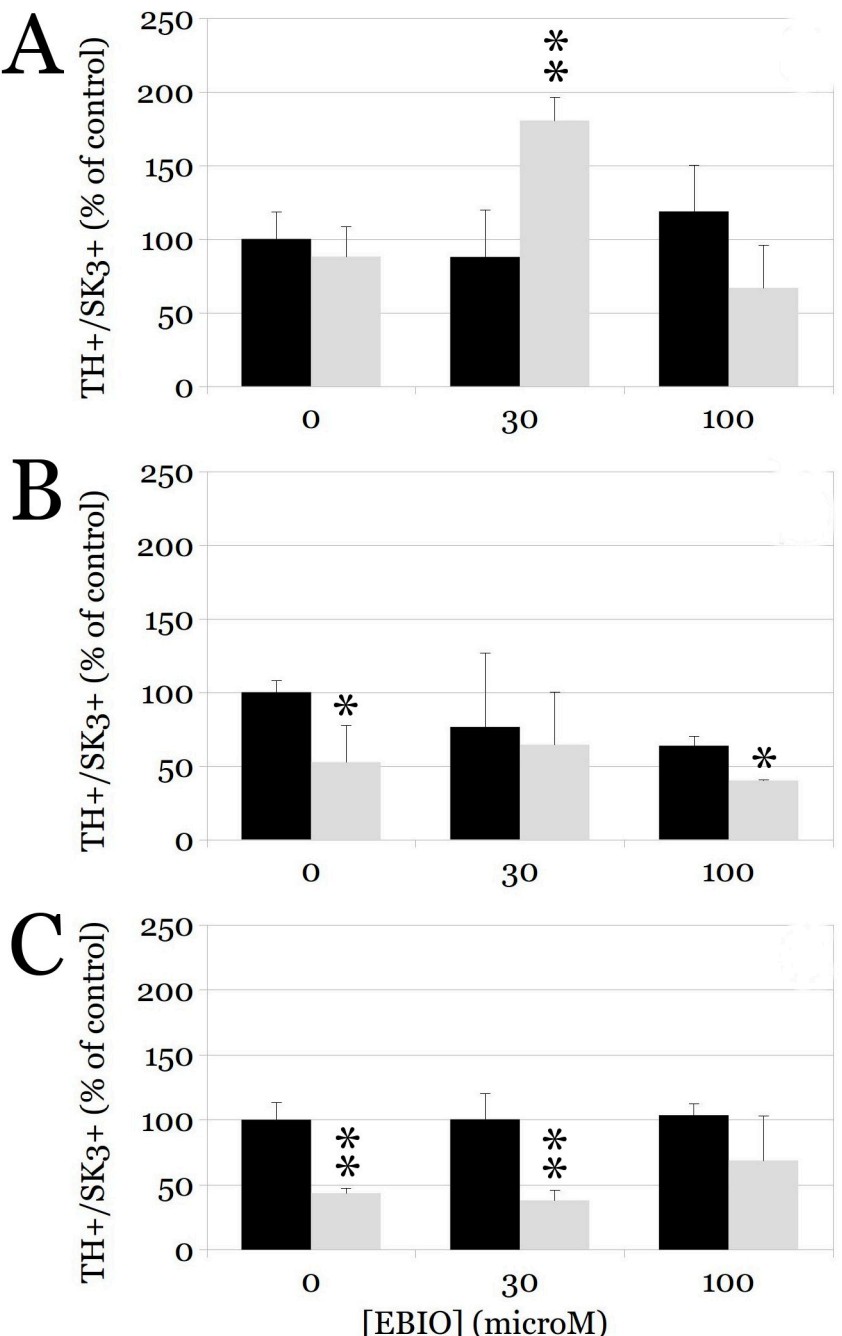

**Fig 4. Concentration-response curves for 1-EBIO effect on DN survival in organotypic cultures.** Bars represent the number of TH+/SK3+ neurons (expressed as % of the control condition) at DIV 8 (panel A), DIV 15 (panel B), and DIV 22 (panel C). Organotypic cultures were treated with conditioned media (control condition, black bars) or with AMPA (100 μM, gray bars) for 16 h. The y axes represent increasing concentrations of 1-EBIO (0, 30 or 100μM). DN survival is expressed as percentage of TH+/SK3+ neurons in each treatment group relative to the control. Data are expressed as mean ± SEM, and are normalized with respect to controls.

synaptic expression of AMPA receptors are tightly regulated [3]. During development in rodents, the four subunits of AMPA receptors (GlurA to GlurD) are first detected in VM neurons at E13, and thereafter their expression increases until roughly E17 [24].

The molecular pathway of AMPA-mediated excitotoxicity in embryonic DN requires activation of voltage-dependent calcium channels, destabilization of calcium homeostasis, activation of a mitochondrial transition pore and nuclear translocation of nuclear factor kappa b, and phosphorylation of p53, this then leads to the activation of a programmed cell death process [18, 19, 38]. However, the key step prior to commitment to die is the suppression of the SK3 current in DN, such that the entire process can be mimicked by the SK3 antagonist apamin, and prevented by SK3 activation by either 1-EBIO or CyPPA (N-cyclohexyl-N-[2-(3,5-dimethyl-pyrazol-1-yl)-6-methyl-4-pyrimidinamine) [3]. Thus, the natural molecular target for neuroprotection in a more intact system is the SK3 channel. Our data is also consistent with the neuroprotective effect of SK channel activation in cultured human postmitotic dopaminergic neurons in vitro following rotenone treatment; indeed, sub-lethal concentrations of rotenone are significantly more toxic to DN and cause more motor impairment in older when compared to younger drosophila flies [37]. Additionally susceptibility of differentiated DN from the human neuroblastoma cell line SH-SY5Y is also increased by time *in vitro* [37].

We found that the effects of 1-EBIO on DN survival following excitotoxicity were complex, but largely consistent with the expected neuroprotection. The contrasting results between our current experiments and those reported by us previously in primary cultures of VM, may be due to differences in culture systems, including the presence of SK3-expressing microglia, which is particularly abundant in the midbrain at this developmental stage [3, 23, 39–41]. Interestingly, SK3 activation in primary cultures led to an increase in TH-expressing neurons, in the absence of any susceptibility to AMPA-induced toxicity [3]; this suggests that SK3 channel expression precedes the ability of DN to trigger the programmed cell death induced by AMPA [19, 21]. At DIV 8, exposure to increasing concentrations of 1-EBIO yielded an inverted U-shaped curve, with an optimum effect at 30 μM and no change in the number of TH+/SK3+ neurons after exposure to 100 μM (Fig 4, panels A-C). The lack of effect at the higher concentration was also seen at DIV15, where 30 μM was clearly neuroprotective but 100 μM had no effect.

On the other hand, at DIV22 exposure of the cultures to 30 μM 1-EBIO did not afford any protection, whereas 100 μM did, suggesting a shift to the right in the concentration-response curve. Again, this may be explained at least in part by the maturational stage of the DN, and possibly also by the indirect effects of 1-EBIO on SK3 expressing microglia or non-dopaminergic neurons, both of which are much more infrequent at DIV22 (Fig 1). Furthermore, SK3 mRNA in the rat brain was found exclusively in areas that also contained large numbers of DA neurons including the substantia nigra (SN) and the ventral tegmental area (VTA), and younger animals express higher levels and less regional variation in SK3 transcripts [42, 43]. As we found, fully differentiated DN continue to express high levels of SK3 channels, which are heavily expressed in the soma and, to a lesser extent, throughout the dendritic arbor [38, 44]. Somatodendritic expression of SK3 channels is consistent with their involvement in pace making activity in DNs, which is also modulated during early development but stable after a week after birth in the rat [45].

A note is needed regarding expression of SK3 immunoreactivity in cells other than DN, which again was most apparent at earlier developmental stages (Fig 1). Since the focus of this publication is the effect of 1-EBIO on the susceptibility of DN we did not attempt to characterize the SK3 expressing cells, but such transient expression has been reported in mesencephalic microglia [39, 46]. Indeed, SK3 channels can modulate inflammatory responses in microglial cells, and SK3 channel blockade inhibited microglial activation and reduced their ability to kill neurons [39, 46]. Thus, it is possible that the presence of microglia in large numbers may alter the overall impact of SK3 activation on excitotoxicity, and that the effect may disappear when microglial numbers are smaller.

Our observations are consistent with findings of the effects of SK3 channel agonists in intact animals [31, 38, 47, 48]. In conditionally SK3-deficient mice, baseline levels of striatal extracellular dopamine are increased [31] while both CyPPA spontaneous firing rate and dopamine release are decreased [49]. As a result, the duration of the apamin-sensitivity afterhyperpolarization is increased and causes an activity-dependent inhibition of current-evoked action potentials in DN [47]. After bilateral lesions with 6-hydroxydopamine into the striatum, SK3 channel expression is reduced in the SN, and motor impairment is reversed by administration of CyPPA [48]. Likewise, inhibition of mitochondrial complex I with rotenone disrupts the dendritic network of human DN and induces neuronal death, and SK3 channel activation preserves the dendritic network, cell viability and ATP levels after rotenone challenge [38].

*KCNN3*, the gene that encodes SK3, is mutated in some families in association with chronic psychosis [4]. Consistent with the results using AMPA excitotoxicity, transgenic expression of the human disease-related SK3 mutation (hSK3Δ) in DN of mice suppresses endogenous SK3 currents, reducing coupling between SK channels and NMDA receptors (NMDARs) and resulting in increased burst firing [4]. Also, hSK3Δ increased excitability of DN through increased evoked calcium signals and potentiated evoked dopamine release [4]. Notably, in mice transgenic for the causal gene for human Huntington Disease, DN excitability was increased due to functional loss and abnormal distribution of SK3 channels. Progressive expansion in the CAG repeat number and nuclear localization of mutant protein within the SN pars compacta resulted in an increase in dopamine release related to a loss of SK3 channel function [50].

## Conclusion

We found that in organotypic cultures of VM, dopaminergic neurons are increasingly susceptible to excitotoxicity with time in vitro, and in direct proportion to their soma size and neurite complexity. Just as we had shown in primary cultures of VM, in orgaontypic cultures treatment with an agonist of SK3 channels results in neuroprotection against excitotoxicity. Notably, we found that the protective effect of SK3 agonists is modulated by age in vitro as well, such that it is most apparent in more mature DN. Our data suggest that SK3 agonists may be a potentially useful target for neuroprotection of developing DN.

## Author Contributions

**Conceptualization:** Gabriel A. de Erausquin.

**Data curation:** Gabriel A. de Erausquin.

**Formal analysis:** Helen M. Belalcazar, Helena Hernandez-Cuervo, Giannina Ladaga, Lucia Padilla, Gabriel A. de Erausquin.

**Funding acquisition:** Gabriel A. de Erausquin.

**Investigation:** Oscar Maldonado, Alexandra Jenkins, Helen M. Belalcazar, Helena Hernandez-Cuervo, Katelynn M. Hyman, Gabriel A. de Erausquin.

**Methodology:** Alexandra Jenkins, Helen M. Belalcazar, Helena Hernandez-Cuervo, Gabriel A. de Erausquin.

**Project administration:** Gabriel A. de Erausquin.

**Resources:** Gabriel A. de Erausquin.

**Writing – original draft:** Oscar Maldonado, Alexandra Jenkins.

**Writing – review & editing:** Helen M. Belalcazar, Helena Hernandez-Cuervo, Katelynn M. Hyman, Giannina Ladaga, Lucia Padilla, Gabriel A. de Erausquin.

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
