## [Decision Letter · Decision Letter 0]

3 Dec 2019

PONE-D-19-26792

Age-dependent neuroprotective effect of an SK3. channel agonist on excitotoxity

to dopaminergic neurons in organotypic culture

PLOS ONE

Dear Dr de Erausquin,

Thank you for submitting your manuscript to PLOS ONE. After careful consideration, we feel that it has merit but does not fully meet PLOS ONE’s publication criteria as it currently stands. Therefore, we invite you to submit a revised version of the manuscript that addresses the points raised during the review process.

We would appreciate receiving your revised manuscript by Jan 17 2020 11:59PM. To enhance the reproducibility of your results, we recommend that if applicable you deposit your laboratory protocols in protocols.io, where a protocol can be assigned its own identifier (DOI) such that it can be cited independently in the future. For instructions see: http://journals.plos.org/plosone/s/submission-guidelines#loc-laboratory-protocols

We look forward to receiving your revised manuscript.

Kind regards,

Faramarz Dehghani

Academic Editor

PLOS ONE

Journal Requirements:

'This work was supported in part by the Center for Brain Health grant from the Valley Baptist

Legacy Foundation to GdE.'

'The funders had no role in study design, data collection and analysis, decision to

publish, or preparation of the manuscript.'

3. Please amend the manuscript submission data (via Edit Submission) to include author Katelynn M. Hyman

Additional Editor Comments (if provided):

Dear Gabriel,

I have now received all comments of the reviewers. As you can see, there is a high number of requests and suggestions made by the reviewers. They are likely to require significant additional work.

If you feel you are able to deal satisfactorily with the comments raised I would be happy to consider a revised version of your work.

Best regards,

Faramarz Dehghani

Reviewers' comments:

Reviewer's Responses to Questions

**Comments to the Author**

1. Is the manuscript technically sound, and do the data support the conclusions?

Reviewer #1: Yes

Reviewer #2: Partly

Reviewer #3: Yes

2. Has the statistical analysis been performed appropriately and rigorously? 

Reviewer #1: Yes

Reviewer #2: Yes

Reviewer #3: Yes

3. Have the authors made all data underlying the findings in their manuscript fully available?

Reviewer #1: Yes

Reviewer #2: Yes

Reviewer #3: No

4. Is the manuscript presented in an intelligible fashion and written in standard English?

Reviewer #1: No

Reviewer #2: No

Reviewer #3: Yes

5. Review Comments to the Author

Reviewer #1: In the present study the authors examined the age-dependency of the neuroprotective effect of an small-conductance calcium activated potassium channel 3 (SK3) agonist, 1-EBIO, on AMPA excitoxicity to dopaminergic neurons (DN) in organotypic VM cultures.

These results indicate that susceptibility of DN to AMPA excitotoxicity is dependent on developmental stag in embryonic VM organotypic cultures, as well as the effects of the SK3 channel agonist 1-EBIO on the survival of SK3-expressing dopaminergic neurons were concentration-dependent and influenced by neuronal developmental stage (DIVs). In all, the data suggest that activation of SK3 channels in DN may be a potentially useful target for neuroprotection.

General impression: this manuscript is interesting and definitely provides the new insight nto the age-dependent neuroprotective effect of an SK3 channel agonist on excitotoxity to dopaminergic neurons in organotypic culture.

However, these manuscript need to be substantial improve and I have numerous remarks that have to be considered:

General points:

The layout of the manuscript is absolutely not according to PLOS ONE. Please do it according to PLOS ONE.

Please say in the whole manuscript “SK3” (without point) or “SK3.” Or is there an interlectual reason to use two versions?

Please change/transfer all references/citations in the manuscript to numbers according to References-List at the end of the manuscript.

Please change the References-List at the end of the manuscript in accordance to PLOS ONE.

Please check the MS for obviously missing word - some sentences do no really give sense as they stand now, eg. last sentence on page 3, first sentence on page 4, last sentence of page 11, ...

Do not say cell culture - you studied organotypic VM cultures.

At page 5 you clearly defined the groups tested. Please stick to these definitions throughout the text - otherwise it is confusing.

Special points:

Abbreviation: Please add all abbreviation used in the manuscript into “Abbreviations” (page 1). Please add also to abbreviations: VM; KCNN3; SK; NMDAR and so on..

Abstract

*Line/row 2: please correct to “calcium-activated”; please write out: 1-EBIO and AMPA.

Line 3: “TH” – please write out.

Line 8 and 12: please use usual and typical mathematic symbols when expressing levels of significance. Please check and correct the similar symbols in the whole manuscript.

Line 9: “VM” – please write out.

Keywords:

Please also add to keywords: “neuroprotective effect”.

Introduction:

Line/row 5: please write “KCNN3” in “italic” for genes.

Line 10: … (AMPA)-mediated excitotoxicity (?); - please add references after these conversation.

Line 20: However, primary cultures are limited in their viability in vitro and only allow observation of short term effects of toxicity (.???.): please add references.

Line 23: Organotypic cultures that retain some of the 3D shape and some modicum of connections with normal anatomical targets, and most importantly, can be maintained in culture for much longer periods of time (.???.) – please add references.

The Introduction should be improve!

Please show more exactly the reason and justify, why you chosen this study design. Please add more description and information about the similar studies in the literature up to date? Please add whole this information into the introduction before: .. .Line 7: Thus, we have focused on understanding the determinants of DN susceptibility to excitotoxic challenges during development.

Materials and Methods:

Please delete “-“at the beginning of the each parts of the Materials and methods.

Pharmacological treatmens

Line 14: .. and AMPA in water as indicated. Please add: which kind of water and more information about this indication (???.).

Stereology and assessment of DN survival

Every other section of VM were stained with TH, and used for unbiased stereological cell counting using the optical fractionator method – please add references to this method.

Counts of TH+ neurons within a counting frame (50% screen height 2) were made at regular predetermined intervals (frame spacing=150 μm) by means of a grid program (Stereologer 2000) - please add the exactly information about this program.

Statistical analysis

.. “by protected post-hoc test for”: please correct to “Post-hoc-Test.

Results:

Don’t use the numbers“1,2,3” before each part of the Results.

2. DNs in organotypic cultures are differentially susceptible to AMPA-receptor mediated excitotoxicity: Please indicate every time very exactly after the Fig. 2 and Fig. 3, wheater it is A or B!! The reader is not a detective for finding the right subfigure.

Please add and correct:

p=0.127, Fig.2????).

to CT tissue (n=15, p=0.005, Fig.2???).

more than half (n=15, p=0.0000000, Fig.2???).

CT tissue (nCT=163,nAMPA=126, p=0.039, Fig. 3 ???).

This pattern was most dramatic at DIV 15 (nCT=151, nAMPA=146, p=2.0 x 10-24, Fig. 3???) and continued to be significant at DIV 22 (nCT=133, nAMPA=111, p=1.7 x10-10, Fig. 3????).

It would be a good idea to write down the exact number and not only the result of the comparative statistics. From the figures the exact values and the SDs or SEMs cannot be seen.

3. Protection conferred by SK3. channel activation is developmental stage-dependent in organotypic cultures of embryonic VM:

The effects of the SK3 channel agonist 1-EBIO on the survival of SK3-expressing dopaminergic neurons were concentration-dependent and influenced by developmental age (Fig. 4, ??? Which panel do you mean??). It would be a good idea to write down the exact number and not only the result of the comparative statistics. From the figures the exact values and the SDs or SEMs cannot be seen. Also say exactly to the previously defined groups.

Discussion:

Please use sometimes instead of “indeed".

Please correct: Perhaps not surprisingly, we found that in organotypic cultures, susceptibility of DN to AMPA-mediated excitotoxicity increases with age and differentiation status, such that the more complex, larger DN that become more common as the tissue matures, are much more susceptible than the embryonic-appearing, small and round neurons more common earlier in development (Fig. 3??? ). A or B?? Please add exactly.

Please add the references after these sentence: We found that the effects of 1-EBIO on DN survival following excitotoxicity were complex, but largely consistent with the expected neuroprotection. The contrasting results between our current and previous experiments (…?? please add the references). What are the previous ones ?

Did you mentioned these results also in the Results part of the manuscript? Interestingly, SK3 activation in younger cultures led to an increase in TH expressing neurons, in the absence of any susceptibility to AMPA-induced toxicity; this suggests that SK3 channel expression precedes the ability of DN to trigger the programmed cell death induced by AMPA (de Erausquin et al., 2003; Dorsey et al., 2006). Indeed, at DIV exposure to increasing concentrations of 1-EBIO yielded an inverted U-shaped curve, with an optimum effect at 30 μM and no change in the number of TH+/SK3+ neurons after exposure to 100 μM. The lack of effect at the higher concentration was also seen at DIV15, where 30 μM was clearly neuroprotective but 100 μM had no effect. On the other hand, at DIV22 exposure of the cultures to 30 μM 1-EBIO did not afford any protection, whereas 100 μM did, suggesting a shift to the right in the concentration response curve.

KCNN3, the gene that encodes SK3. is mutated in some families in association with chronic psychosis (Soden et al., 2013): Please write “KCNN3” in “italic” for gene.

Please come up with a conclusio at the ned.

Figure Legends:

Please write in the Fig. 2 and Fig. 3: (A), (B) instead of A, B.

Please use in all Figures: ± SEM instead of +/- SEM.

Figure 4: The Legend should be rewritten!! It is not understandable and it is not possible to see in the Fig. 4 the different incubation conditions: Cultures were incubated with 1-EBIO alone (30-100μM), AMPA alone (100 μM), or 1- EBIO (30-100μM) plus AMPA (100 μM) for 16 hr at DIV 8, DIV 15, and DIV 22. Please include all these conditions into layout of the Fig. 4.

Supplementary Fig. 1: Please describe the legend in full, not refering to <fig 4=""></fig>

<fig 4="">

Reviewer #2: The manuscript describes the protective effects of a SK3 enhancer in an organotypic culture model of AMPA toxicity.

The experimental model is very artifactual. If it can be used to study basic, molecular properties of neurons that can endure dissection and long-term culturing, it is questionable whether this in vitro system is able to model normal neurodevelopment and associated pathological conditions.

Materials and methods are sufficiently clear and detailed.

Results are very limited and sometimes contradictory.

One example: why 1-EBIO is protective at 30 and not 100 uM at one stage and the other way at later stage?

Also, the paper suffers from a number of limits that make it unsuitable for publication at this stage.

1) the quality of language, both in terms of english form and logic presentation of concepts is very poor.

2) the abstract lacks an introductory section which is normally required for the reader to understand the aim of the study.

3) It is unclear the rationale linkfing AMPA toxicity and neurological disorder such as schizophrenia as, according to the current comprehension of the disease, there is no neurodevelopmental neuronal loss linked to the disease.

4) the quality of images in immunostaining figures is not acceptable.

5) the SK3 staining shows very high background. I find it hard to believe that the signal is specific.

The discussion is too long for such a scarce amount of data. Finally, it contains some useful background information that should be moved forward into the intro.

Reviewer #3: In this paper, Authors aimed to investigate age-dependent neuroprotective effects of a conductance calcium activated potassium channel 3 (SK3) agonist, 1-EBIO, on AMPA excitoxicity to dopaminergic neurons (DN) in organotypic cultures. They claimed that exposure of organotypic cultures to AMPA (100 μm, 16 h) had no effect on the survival of DN+ at DIV 8, but caused significant toxicity at DIV 15. Moreover, immature DN+ (small, bipolar) were increased after AMPA (100 μm, 16 h) at DIV 8. This effect was more significant at DIV 15 and at DIV 22. Administration of 30 μM 1-EBIO resulted in a large increase in DN+ at DIV 8 and in AMPA toxicity prevention at DIV 15, while no effects on excitotoxicity were observed at DIV 22. Different results were observed with administarion of 100 um of the same compound, therefore suggesting that its effects were concentration-dependent and influenced by neuronal developmental stage.

I found this paper interesting. However, there are some major concerns that Authors should address before it can be reconsidered for publication:

1. In the Introduction section, the general and specific aims of the study should be better presented.

2. References should be provided for the dissection procedure (Materials and Methods section).

3. The same as above for the description of cell culture (Materials and Methods section).

4. References should be also provided for the doses of the pharmacological treatment used in this study.

5. For section Immunohistochemistry and Immunofluorence, Authors should detail the validation process of the antibody they used, especially with respect to the use of negative controls.

6. References should be provided for section Stereology and assessment of DN survival.

7. Authors should include the F values obtained for each performed ANOVA (in the description of results or in the figure legends).

8. The graphical quality of Images include in Fig.1 should be improved.

9. The quality of Figure 3 should be improved, especially for panel B.

10. I found the discussion quite fragmented and, in some of its part, a repetition of the obtained results. Therefore, Authors should revise this section, better discussing their results, also in the light of the existing literature in the field, better elaborating on the novelty of their findings.</fig>

<fig 4="">**********

6. PLOS authors have the option to publish the peer review history of their article (what does this mean?). If published, this will include your full peer review and any attached files.

Reviewer #1: Yes: Andreas Wree

Reviewer #2: No

Reviewer #3: No

</fig>

---

## [Author Response · Author response to Decision Letter 0]

3 Apr 2020

Please find our replies to the comments in the attached file.

---

## [Decision Letter · Decision Letter 1]

17 Apr 2020

PONE-D-19-26792R1

Age-dependent neuroprotective effect of an SK3 channel agonist on excitotoxity to dopaminergic neurons in organotypic culture

PLOS ONE

Dear Dr de Erausquin,

Thank you for submitting your manuscript to PLOS ONE. After careful consideration, we feel that it has merit but does not fully meet PLOS ONE’s publication criteria as it currently stands. Therefore, we invite you to submit a revised version of the manuscript that addresses the points raised during the review process.

We would appreciate receiving your revised manuscript by Jun 01 2020 11:59PM. To enhance the reproducibility of your results, we recommend that if applicable you deposit your laboratory protocols in protocols.io, where a protocol can be assigned its own identifier (DOI) such that it can be cited independently in the future. For instructions see: http://journals.plos.org/plosone/s/submission-guidelines#loc-laboratory-protocols

We look forward to receiving your revised manuscript.

Kind regards,

Faramarz Dehghani

Academic Editor

PLOS ONE

Additional Editor Comments (if provided):

Dear Dr. de Erausquin,

I have now received the comments from the reviewers on the revised version of your manuscript. Since reviewer 3 was satisfied with the changes made both other reviewers still have substantial concerns.

You will find the detailed comments below. I share the points of criticism but encourage you to address all points accordingly and thoroughly revise the manuscript.

Best regards,

Faramarz Dehghani

Reviewers' comments:

Reviewer's Responses to Questions

**Comments to the Author**

1. If the authors have adequately addressed your comments raised in a previous round of review and you feel that this manuscript is now acceptable for publication, you may indicate that here to bypass the “Comments to the Author” section, enter your conflict of interest statement in the “Confidential to Editor” section, and submit your "Accept" recommendation.

Reviewer #1: (No Response)

Reviewer #2: (No Response)

Reviewer #3: All comments have been addressed

2. Is the manuscript technically sound, and do the data support the conclusions?

Reviewer #1: No

Reviewer #2: No

Reviewer #3: Yes

3. Has the statistical analysis been performed appropriately and rigorously? 

Reviewer #1: I Don't Know

Reviewer #2: Yes

Reviewer #3: Yes

4. Have the authors made all data underlying the findings in their manuscript fully available?

Reviewer #1: No

Reviewer #2: Yes

Reviewer #3: Yes

5. Is the manuscript presented in an intelligible fashion and written in standard English?

Reviewer #1: No

Reviewer #2: No

Reviewer #3: Yes

6. Review Comments to the Author

Reviewer #1: This manuscript was substantively improved after revision.

The main massage of the study became substabtionally clearer.

Unfortunately, authors provided not all considerable corrections and therefore, this manuscript should be further improved and corrected before publishing.

There are still a lot of many "minors":

- Minor spell check required.

- Please indicate all references according to PlosOne.

- Please say “SK3” and check and correct once again in whole manuscript.

One example in Abstract: “The effects of the SK3. channel agonist 1-EBIO on the survival of SK3.- expressing dopaminergic neurons were concentration-dependent and influenced by neuronal developmental stage”. one more: in Results section: “SK3. channels are expressed in DN organotypic cultures”.

- Abstract: Please structure your Abstract section according to PlosOne: The Abstract section should include a separate parts: Background; Methods; Results; Conclusion. Already before, it was a proposal from me to design your manuscript according to PlosOne.

- Introduction: The Introduction section is still too short to describe the situation of the appropriate literature and the results of the previous publications. The Introduction section should be once again improve. Please describe more exactly why you selected an AMPA-model for your study?

- Please add more exactly the results of the Reference 3. Please describe more exactly and add to Introduction the description and the results of the References 10, 14,15.

- Pharmacological treatments: You said: “Cultures used for SK3 channel activation experiments were exposed to 1-ethyl-2-benzimidazolinone (1-EBIO) (30, 100 μM), and (AMPA) (100 μM) for 16 hours (DIV 7, 14, 21) in the following treatment groups: 1-EBIO 30 μM, 1-EBIO 100 μM, 1-EBIO 30 μM plus AMPA 100 μM, 1-EBIO 100 μM plus AMPA 100 μM, and AMPA 100 μM”.

Please add, why you used this appropriate doses and volumes?

- Protection conferred by SK3 channel activation is developmental stage-dependent in organotypic cultures of embryonic VM: Please say: (Fig. 4, panel A); (Fig. 4, panel B); (Fig. 4, panel C) like as above. Please use the same abbreviations style in whole text, all Figures and Legends.

- Discussion: Please add more references at the end of this sentence:” Dopamine neuron susceptibility to excitotoxic injury is potentially relevant to a number of pathological conditions, and therefore so is the development of potential neuroprotective strategies, including the use of pharmacological modulators of SK3 channel function”.

- You said: “This is consistent with observations in other models, including susceptibility to rotenone-induced toxicity (21). Please add more exactly description of the study and results of the Reference 21.

- You said: “Overall, AMPA did not induce cell death in DN at 8 days in vitro, but toxicity became apparent at 15 DIV and most intense at 22 DIV”. Please add to discussion, what you think about that. This - I think - is very essential in the message of the study.

- You said: “We previously showed that 1-EBIO effectively protects DN in primary cultures of ventral mesencephalon against AMPA-mediated toxicity by sustaining SK3 channel conductance (3). Please add more exactly description of the study and results of the Reference 3.

- You said:” Our data are also consistent with the neuroprotective effect of SK channel activation in cultured human postmitotic dopaminergic neurons in vitro following rotenone treatment (23). Please add more exactly description of the study and results of the Reference 23.

- You said: “The contrasting results between our current experiments and those reported by us previously in primary cultures of VM (3)”. Please add more exactly description of the study and results of the Reference 3.

- Please add more references at the end of this sentence: “Our observations are consistent with findings of the effects of SK3 channel agonists in intact animals“.

- Conclusion

The Conclusion section is too short. Please add the meaning of your method and your findings for experimental research and clinical practice. Please add also to Conclusion section the future perspectives.

- Figures: The quality of Figures 1 and 3 is still not acceptable for publishing. They should be substantial once more improved.

- Figure 1: Please indicate the panels A, B and C directly in your Figure 1.

Please indicate the DIV8, DIV15 and DIV22 directly in your Figure 1, like as Figures 2 and 3.

- Figure 4: Please indicate the panels A, B and C directly in your Figure 4.

Figure Legends:

Fig. 1:

Please indicate the panels instead of a, b, c like as A, B, C, the same with text and images and in other Legends.

Reviewer #2: The manuscript entitled "Age-dependent neuroprotective effect of an SK3 channel agonist on excitotoxity to dopaminergic neurons in organotypic culture" was, in the initial version, unsuited for publication due to, primarily, an unacceptable number of typos, imprecisions and omissions in the text and general lack of accuracy.

The modest improvements made in this resubmission are by all means insufficient. In fact, the manuscript is still flawed by many of the serious shortcomings that all reviewers pointed out in the previous version.

If such generalized untidiness is regrettable in a first submission, I find it intolerable in a resubmission.

A) Major issues

Abstract: line 1, “an small conductance (..a small…)

Introduction:

1) Line 2, a bracket is missing in the citation.

2) Line 5, “…risk of schizophrenia in some samples…”. Maybe the author meant “examples”?

3) Line 6, “To further understanding of the determinants” (…understand the…).

4) Line 16, a number is missing in the citation in brackets.

5) Line 18, “Organotypic cultures that…”. “That” should be removed.

Results

1) Title: “SK3. channels…”. The abbreviation SK3 is still “SK3.” here and in many spots throughout the text. Again, this was incomprehensible in the first version, it is unacceptable after this issue was explicitly raised by reviewers.

2) 1st paragraph, line 4, “panel A”. There is no panel A, B or C in figure 1, only three pictures in a left to right row.

3) Page 11, line 3: AMPA-receptor-mediated. There is one too many hyphens.

4) Page 11, line 4, “DIV8 treated…”. Treated at DIV8 would be more correct.

5) Page 13, line 16, “…leading activation…” (…leading to the activation…)

6) Page 14, line 9, “At DIV…”. 8? 15? 22? Please specify.

7) Page 16, Conclusion. The section was clearly added simply to meet the journal required formta, but a 2-line period is not acceptable.

8) Acknowledgements: if, as I understand, no acknowledgements are to be made, please insert at least one sentence stating this.

References: SK3. is abundantly present throughout.

Legends and relative figures

1) Figure 1. There are no A, B and C letters in the figure. The resolution of the pictures is very low, so that individual, large pixels can easily be distinguished in paper and on screen. What is the vertical “background” writing on the right? Last, but not least, there is no scale bar in the photographs, although it is mentioned at the end of the legend.

2) Figure 3, the resolution is insufficient, the appearance of the text is very blurred in both panels A and B

3) Figure 4. There are no A, B or C letters in the panels of figure 4. Line 4 of legend, …the y axes…(They are actually the x axes).

B) Minor issues: beyond form, that makes it unacceptable, the MS still suffers from many of the scientific and conceptual shortcomings raised by reviewers.

1) Abstract. The abstract was minimally revised. In fact, it still lacks an overview of the field and the scientific question the MS aims to address. This is indispensable in the abstract of a scientific publication. Instead, the 2nd sentence reports an unnecessary characterization of the model system.

2) Discussion. Page 13, second paragraph; Stage-dependent susceptibility to AMPA toxicity, the underpinnings of which are not discussed in any detail here, is likely due to progressive maturation of excitatory synapses and/or intracellular AMPA signals. Susceptibility to rotenone, a mitochondrial poison, does not seem a useful example of similarity. Rather, I find it confounding.

3) Page 13-14. Neuroprotection afforded by 1-EBIO (only hypothetically via activation of SK3 channels as there is no direct demonstration of K current enhancement in the model) is indeed complex, not to say hardly intelligible. One result stands out that has been inadequately discussed and that conflicts with the authors interpretation of the link between AMPA toxicity and 1-EBIO-dependent neuroprotection: 30 uM 1-EBIO increases the number of TH+/SK3+ neurons at DIV 8 in the absence of toxic AMPA effects (fig 4, top). This suggests a neurotrophic action of 1-EBIO treatment, at least at this stage. Consistently, the neuroprotection afforded 1-EBIO in other conditions may be, at least in part, explained by this effect and, thus, not solely a counteraction of AMPA activation/signaling.

Reviewer #3: (No Response)

7. PLOS authors have the option to publish the peer review history of their article (what does this mean?). If published, this will include your full peer review and any attached files.

Reviewer #1: No

Reviewer #2: No

Reviewer #3: No

---

## [Author Response · Author response to Decision Letter 1]

3 May 2020

Comment: “The main massage of the study became substantially clearer.”

Reply: We appreciate the encouragement.

Comment: Unfortunately, authors provided not all considerable corrections and therefore, this manuscript should be further improved and corrected before publishing. There are still a lot of many "minors"

Reply: We will address them individually.

Comment: Minor spell check required.

Reply: We thoroughly spell checked the manuscript.

Comment: Please indicate all references according to PlosOne.

Reply: References were individually and manually edited according to PlOS One guidelines.

Comment: Please say “SK3” and check and correct once again in whole manuscript.

Reply: We have corrected the abbreviation where needed.

Comment: Abstract: Please structure your Abstract section according to PlosOne: The Abstract section should include a separate parts: Background; Methods; Results; Conclusion. Already before, it was a proposal from me to design your manuscript according to PlosOne.

Reply: We apologize about missing the comment the first time. We have formatted the Abstract as required.

Comment: Introduction: The Introduction section is still too short to describe the situation of the appropriate literature and the results of the previous publications. The Introduction section should be once again improved. Please describe more exactly why you selected an AMPA-model for your study?

Reply: As requested, we have expanded the rationale for the use of this model. Of note, the laboratory of the corresponding author established this model two decades ago, and has contributed significantly to the literature on the topic.

Comment: Please add more exactly the results of the Reference 3. Please describe more exactly and add to Introduction the description and the results of the References 10, 14,15.

Reply: We respectfully disagree. The present manuscript is not a review. The requested results are already sufficiently described in great detail in the referenced manuscripts by our own group, and any reader can satisfy their questions by seeking out those references. Further repetition here would detract from the main purpose of this original publication.

Comment: Pharmacological treatments: You said: “Cultures used for SK3 channel activation experiments were exposed to 1-ethyl-2-benzimidazolinone (1-EBIO) (30, 100 μM), and (AMPA) (100 μM) for 16 hours (DIV 7, 14, 21) in the following treatment groups: 1-EBIO 30 μM, 1-EBIO 100 μM, 1-EBIO 30 μM plus AMPA 100 μM, 1-EBIO 100 μM plus AMPA 100 μM, and AMPA 100 μM”.

Please add, why you used this appropriate doses and volumes?

Reply: The doses were chosen based on recommendations from manufacturers or previously published data by our own and other relevant groups. We have added this sentence to the manuscript.

Comment: Protection conferred by SK3 channel activation is developmental stage-dependent in organotypic cultures of embryonic VM: Please say: (Fig. 4, panel A); (Fig. 4, panel B); (Fig. 4, panel C) like as above. Please use the same abbreviations style in whole text, all Figures and Legends.

Reply: We corrected the requested format.

Comment: Discussion: Please add more references at the end of this sentence:” Dopamine neuron susceptibility to excitotoxic injury is potentially relevant to a number of pathological conditions, and therefore so is the development of potential neuroprotective strategies, including the use of pharmacological modulators of SK3 channel function”.

Reply: We added three references as requested.

Comment: You said: “This is consistent with observations in other models, including susceptibility to rotenone-induced toxicity (21). Please add more exactly description of the study and results of the Reference 21.

Reply: The relevant findings described in reference 21 have been added to the discussion text.

Comment: You said: “Overall, AMPA did not induce cell death in DN at 8 days in vitro, but toxicity became apparent at 15 DIV and most intense at 22 DIV”. Please add to discussion, what you think about that. This - I think - is very essential in the message of the study.

Reply: We have added two paragraphs to the discussion to address this comment.

Comment: You said: “We previously showed that 1-EBIO effectively protects DN in primary cultures of ventral mesencephalon against AMPA-mediated toxicity by sustaining SK3 channel conductance (3). Please add more exactly description of the study and results of the Reference 3.

Reply: We have added several paragraphs expounding the findings of our previous publication (reference 3).

Comment: You said:” Our data are also consistent with the neuroprotective effect of SK channel activation in cultured human postmitotic dopaminergic neurons in vitro following rotenone treatment (23). Please add more exactly description of the study and results of the Reference 23.

Reply: We have added a paragraph with the description of the study referenced.

Comment: You said: “The contrasting results between our current experiments and those reported by us previously in primary cultures of VM (3)”. Please add more exactly description of the study and results of the Reference 3.

Reply: The results from our previous publication are now extensively discussed in previous paragraphs.

Comment: Please add more references at the end of this sentence: “Our observations are consistent with findings of the effects of SK3 channel agonists in intact animals“.

Reply: All of the available studies are referenced. We regret to say that no other data are available, but this is why our paper is relevant…

Comment: The Conclusion section is too short. Please add the meaning of your method and your findings for experimental research and clinical practice. Please add also to Conclusion section the future perspectives.

Reply: We have expanded the conclusion as requested.

Comment: Figures: The quality of Figures 1 and 3 is still not acceptable for publishing. They should be substantial once more improved.

Reply: The figures are very high quality and passed the quality tests by PlOS One. We wonder if the reviewer is looking at the uploaded tiff files or instead referencing the jpg transformations inside the manuscript?

Comments: Figure 1: Please indicate the panels A, B and C directly in your Figure 1. Please indicate the DIV8, DIV15 and DIV22 directly in your Figure 1, like as Figures 2 and 3. Figure 4: Please indicate the panels A, B and C directly in your Figure 4. Figure Legends: Fig. 1: Please indicate the panels instead of a, b, c like as A, B, C, the same with text and images and in other Legends.

Reply: We have corrected the figures and legends as suggested.

Comment: Reviewer #2: The manuscript entitled "Age-dependent neuroprotective effect of an SK3 channel agonist on excitotoxity to dopaminergic neurons in organotypic culture" was, in the initial version, unsuited for publication due to, primarily, an unacceptable number of typos, imprecisions and omissions in the text and general lack of accuracy. The modest improvements made in this resubmission are by all means insufficient. In fact, the manuscript is still flawed by many of the serious shortcomings that all reviewers pointed out in the previous version. If such generalized untidiness is regrettable in a first submission, I find it intolerable in a resubmission.

Reply: Dear Editor, the corresponding author for this manuscript is a PlOS One Academic Editor and reviews frequently for other highly ranked journals. I find the tone and content of this comment unprofessional, inaccurate and offensive. To state that a typo (“an” for “a”) in an abstract, or the lack of a bracket in a citation are “major issues” is flatly outrageous and not in keeping with review guidelines or policy by PlOS One. In spite of this problem, and for the sake of facilitating publication and editorial review, we have corrected each and everyone of the very minor issues raised by this reviewer, and reply to each in the following paragraphs.

Comment: Abstract: line 1, “an small conductance (..a small…)

Reply: The full sentence read: “an small conductance … agonist” and was grammatically correct. Nonetheless, we have changed it to read “an SK3 agonist”, to make it better sounding.

Comment: 1) Line 2, a bracket is missing in the citation. 

Reply: We added the bracket.

Comment: 2) Line 5, “…risk of schizophrenia in some samples…”. Maybe the author meant “examples”?. 

Reply: No, we meant samples of patients with schizophrenia. We corrected it to “patient samples” to clarify the meaning of the sentence.

Comment: 3) Line 6, “To further understanding of the determinants” (…understand the…). 

Reply: The sentence is grammatically correct and uses the gerund “understanding” as a form that is derived from a verb but that functions as a noun.

Comment: 4) Line 16, a number is missing in the citation in brackets.

Reply: We added the missing bracket.

Comment: 5) Line 18, “Organotypic cultures that…”. “That” should be removed.

Reply: The reviewer is correct, and we removed the word “That” from the sentence.

Comment: 1) Title: “SK3. channels…”. The abbreviation SK3 is still “SK3.” here and in many spots throughout the text. Again, this was incomprehensible in the first version, it is unacceptable after this issue was explicitly raised by reviewers.

Reply: We have corrected the usage of SK3 throughout the manuscript.

Comment: 2) 1st paragraph, line 4, “panel A”. There is no panel A, B or C in figure 1, only three pictures in a left to right row.

Reply: We have corrected the figures as requested by reviewer #1.

Comment: 3) Page 11, line 3: AMPA-receptor-mediated. There is one too many hyphens.

Reply: We removed the second hyphen.

Comment: 4) Page 11, line 4, “DIV8 treated…”. The sentence is no longer in the text of the manuscript.

Comment: 5) Page 13, line 16, “…leading activation…” (…leading to the activation…)

Reply: The reviewer is correct and the sentence was changed accordingly.

Comment: 6) Page 14, line 9, “At DIV…”. 8? 15? 22? Please specify.

Reply: It should have read At DIV8. The typo was corrected (thank you).

Comment: 7) Page 16, Conclusion. The section was clearly added simply to meet the journal required formta, but a 2-line period is not acceptable.

Reply: As requested by reviewer #1 also, we have expanded the conclusion.

Comment: 8) Acknowledgements: if, as I understand, no acknowledgements are to be made, please insert at least one sentence stating this.

Reply: We added the requested sentence.

Comment: References: SK3. is abundantly present throughout.

Reply. The usage has been corrected throughout.

Comment: 1) Figure 1. There are no A, B and C letters in the figure. The resolution of the pictures is very low, so that individual, large pixels can easily be distinguished in paper and on screen. What is the vertical “background” writing on the right? Last, but not least, there is no scale bar in the photographs, although it is mentioned at the end of the legend. 2) Figure 3, the resolution is insufficient, the appearance of the text is very blurred in both panels A and B. 3) Figure 4. There are no A, B or C letters in the panels of figure 4. Line 4 of legend, …the y axes… (They are actually the x axes).

Reply: As in the comment from reviewer #1, I wonder if the reviewers are looking at the pdf file instead of at the uploaded tiff individual files? The figures are highest quality and passed the PlOS One quality test. We fixed the other issues raised by the comment.

Comment: 1) Abstract. The abstract was minimally revised. In fact, it still lacks an overview of the field and the scientific question the MS aims to address. This is indispensable in the abstract of a scientific publication. Instead, the 2nd sentence reports an unnecessary characterization of the model system.

Reply: The abstract has been thoroughly re-written.

Comment: 2) Discussion. Page 13, second paragraph; Stage-dependent susceptibility to AMPA toxicity, the underpinnings of which are not discussed in any detail here, is likely due to progressive maturation of excitatory synapses and/or intracellular AMPA signals. Susceptibility to rotenone, a mitochondrial poison, does not seem a useful example of similarity. Rather, I find it confounding.

Reply: This aspect of the discussion has been expanded following suggestions from reviewer #1 as well. We agree with the reviewer on the likely molecular mechanism of the age-dependent effects.

Comment: 3) Page 13-14. Neuroprotection afforded by 1-EBIO (only hypothetically via activation of SK3 channels as there is no direct demonstration of K current enhancement in the model) is indeed complex, not to say hardly intelligible. One result stands out that has been inadequately discussed and that conflicts with the authors interpretation of the link between AMPA toxicity and 1-EBIO-dependent neuroprotection: 30 uM 1-EBIO increases the number of TH+/SK3+ neurons at DIV 8 in the absence of toxic AMPA effects (fig 4, top). This suggests a neurotrophic action of 1-EBIO treatment, at least at this stage. Consistently, the neuroprotection afforded 1-EBIO in other conditions may be, at least in part, explained by this effect and, thus, not solely a counteraction of AMPA activation/signaling.

Reply: The direct effects of AMPA current modulation on dopaminergic neuron survival and on the mechanism of action of SK3 agonists was extensively demonstrated in our previous work (Benitez et al). We have included those results in greater detail in the current version of the discussion.

---

## [Decision Letter · Decision Letter 2]

2 Jun 2020

PONE-D-19-26792R2

Age-dependent neuroprotective effect of an SK3 channel agonist on excitotoxity to dopaminergic neurons in organotypic culture

PLOS ONE

Dear Dr. de Erausquin,

Thank you for submitting your manuscript to PLOS ONE. After careful consideration, we feel that it has merit but does not fully meet PLOS ONE’s publication criteria as it currently stands. Therefore, we invite you to submit a revised version of the manuscript that addresses the points raised during the review process.

We look forward to receiving your revised manuscript.

Kind regards,

Faramarz Dehghani

Academic Editor

PLOS ONE

Reviewers' comments:

Reviewer's Responses to Questions

**Comments to the Author**

1. If the authors have adequately addressed your comments raised in a previous round of review and you feel that this manuscript is now acceptable for publication, you may indicate that here to bypass the “Comments to the Author” section, enter your conflict of interest statement in the “Confidential to Editor” section, and submit your "Accept" recommendation.

Reviewer #1: (No Response)

2. Is the manuscript technically sound, and do the data support the conclusions?

Reviewer #1: Yes

3. Has the statistical analysis been performed appropriately and rigorously? 

Reviewer #1: Yes

4. Have the authors made all data underlying the findings in their manuscript fully available?

Reviewer #1: Yes

5. Is the manuscript presented in an intelligible fashion and written in standard English?

Reviewer #1: Yes

6. Review Comments to the Author

Reviewer #1: This manuscript was very substantively improved after both major revisions.

Unfortunately, the authors provided once again not all considerable corrections and furthermore, some problems emerged acutely during the last revision by authors.

Therefore, this manuscript need once more a bit mprovement and corrections before publishing. I hope, then eberything will fine.

Correction points:

Abstract

Authors said: Background: Small conductance, calcium-activated (SK3) potassium channels control the intrinsic excitability of dopaminergic neurons (DN) in the midbrain and modulate their susceptibility to toxic insults during development.

Please use another sentence, this sentence is absolutely the same with the first sentence in your Introduction section.

Materials and Methods

Animals and Dissections

Authors said: Timed pregnant female Sprague-Dawley rats (Charles River Laboratories, MA, USA) were euthanized by exposure to CO2 on day 14 of gestation.

Please add the total number of pregnant female Sprague-Dawley rats used, (n=?).

Pharmacological treatment

Authors said: All drug concentrations were chosen based on recommendations from manufacturers or previously published data by our own and other relevant groups.

Please add the corresponding references for appropriate previously published data by yourself and others.

Discussion

Authors said: These findings are consistent with knowledge about the expression of AMPA receptors in DN in VM or rodents.

Please add references at the end of this sentence.

Authors said: Additionaly susceptibility of differentiated DN from the human cell line SH-SY5Y is also increased by time in vitro (37).

Please write out: SH-SY5Y. By the way, please correct “Additionaly” to “Additionally”.

Authors said: Our observations are consistent with findings of the effects of SK3 channel agonists in intact animals.

Please add references at the end of this sentence.

Figure 1

Please indicate the DIV8, DIV15 and DIV22 directly in your Figure 1, like as Figures 2 and 3.

Figure 4

Please indicate the panels A, B and C directly in your Figure 4.

Figure Legends

Fig. 1:

Please indicate the panels instead of a, b, c like as A, B, C, the same with text and images and in other Legends.

Please correct two time “anti-SK3.” to “anti-SK3”

Please see both sentences: Images show examples of stained neurons at DIV 8 (panel a), DIV 15 (panel b) and DIV 22 (panel c) merged color images show high co-localization for anti-TH (red) and anti-SK3. (green) antibodies. At each DIV there are also a number of cells that are positive for anti-SK3., but negative for anti-TH. At DIV 8 DNs were typically bipolar with small fusiform shaped somas and displayed little branching.

Fig. 3:

Please delete from the legends: *p < 0.05, **p < 0.01, ***p <0.001. These p values and asterisks were not used in the Figure 3.

7. PLOS authors have the option to publish the peer review history of their article (what does this mean?). If published, this will include your full peer review and any attached files.

Reviewer #1: No

---

## [Author Response · Author response to Decision Letter 2]

5 Jul 2020

Reply to Reviewer’s comments

Comment: “Please use another sentence, this sentence is absolutely the same with the first sentence in your Introduction section”.

Reply: Thank you for pointing this out. We re-wrote the introduction.

Comment: Please add the total number of pregnant female Sprague-Dawley rats used, (n=?).

Reply: we added the requested number

Comment: Please add the corresponding references for appropriate previously published data by yourself and others.

Reply: references numbers were added as requested

Comment:Please add references at the end of this sentence.

Reply: reference number was added as requested

Comment: Please write out: SH-SY5Y. 

Reply: The code proided is the only name used for that cell line, not an abbreviation. There are multiple references (included the one we provide), but see for instance https://www.sigmaaldrich.com/catalog/product/sigma/cb_94030304?lang=en&region=US&gclid=EAIaIQobChMIiJizttmx6gIVEL7ACh10qAQdEAAYASAAEgIbWvD_BwE

We clarified that it is a human neuroblastoma cell line.

Comment: By the way, please correct “Additionaly” to “Additionally”.

Reply: Corrected as noted. Thank you.

Comment: Please add references at the end of this sentence.

Reply: Added the reference as requested.

---

## [Decision Letter · Decision Letter 3]

8 Jul 2020

Age-dependent neuroprotective effect of an SK3 channel agonist on excitotoxity to dopaminergic neurons in organotypic culture

PONE-D-19-26792R3

Dear Dr. de Erausquin,

We’re pleased to inform you that your manuscript has been judged scientifically suitable for publication and will be formally accepted for publication once it meets all outstanding technical requirements.

Kind regards,

Faramarz Dehghani

Academic Editor

PLOS ONE

Additional Editor Comments (optional):

Reviewers' comments:

Reviewer's Responses to Questions

**Comments to the Author**

1. If the authors have adequately addressed your comments raised in a previous round of review and you feel that this manuscript is now acceptable for publication, you may indicate that here to bypass the “Comments to the Author” section, enter your conflict of interest statement in the “Confidential to Editor” section, and submit your "Accept" recommendation.

Reviewer #1: All comments have been addressed

2. Is the manuscript technically sound, and do the data support the conclusions?

Reviewer #1: Yes

3. Has the statistical analysis been performed appropriately and rigorously? 

Reviewer #1: Yes

4. Have the authors made all data underlying the findings in their manuscript fully available?

Reviewer #1: Yes

5. Is the manuscript presented in an intelligible fashion and written in standard English?

Reviewer #1: Yes

6. Review Comments to the Author

Reviewer #1: (No Response)

7. PLOS authors have the option to publish the peer review history of their article (what does this mean?). If published, this will include your full peer review and any attached files.

Reviewer #1: No

---

## [Editor Report · Acceptance letter]

10 Jul 2020

PONE-D-19-26792R3 

Age-dependent neuroprotective effect of an SK3 channel agonist on excitotoxity to dopaminergic neurons in organotypic culture 

Dear Dr. de Erausquin:

I'm pleased to inform you that your manuscript has been deemed suitable for publication in PLOS ONE. Congratulations! Your manuscript is now with our production department. 

Kind regards, 

on behalf of

Dr. Faramarz Dehghani 

Academic Editor

PLOS ONE